# Prediction of Daily Ambient Temperature and Its Hourly Estimation Using Artificial Neural Networks in an Agrometeorological Station in Castile and León, Spain

**DOI:** 10.3390/s22134850

**Published:** 2022-06-27

**Authors:** Francisco J. Diez, Adriana Correa-Guimaraes, Leticia Chico-Santamarta, Andrés Martínez-Rodríguez, Diana A. Murcia-Velasco, Renato Andara, Luis M. Navas-Gracia

**Affiliations:** 1Department of Agricultural and Forestry Engineering, University of Valladolid, Campus La Yutera, 34004 Palencia, Spain; x5pino@yahoo.es (F.J.D.); adriana.correa@uva.es (A.C.-G.); andres.martinez.rodriguez@uva.es (A.M.-R.); dianaalexandra.murcia@alumnos.uva.es (D.A.M.-V.); 2International Department, Harper Adams University, Newport TF10 8NB, UK; lchico-santamarta@harper-adams.ac.uk; 3Directorate of Research and Graduate Studies, Antonio José de Sucre National Experimental Polytechnic University, Barquisimeto 3001, Venezuela; randara@unexpo.edu.ve

**Keywords:** ambient temperature, evapotranspiration, agrometeorology, artificial neural networks (ANNs), precision agriculture, prediction

## Abstract

This study evaluates the predictive modeling of the daily ambient temperature (maximum, T_max_; average, T_ave_; and minimum, T_min_) and its hourly estimation (T_0h_, …, T_23h_) using artificial neural networks (ANNs) for agricultural applications. The data, 2004–2010, were used for training and 2011 for validation, recorded at the SIAR agrometeorological station of Mansilla Mayor (León). ANN models for daily prediction have three neurons in the output layer (T_max_(t + 1), T_ave_(t + 1), T_min_(t + 1)). Two models were evaluated: (1) with three entries (T_max_(t), T_ave_(t), T_min_(t)), and (2) adding the day of the year (J(t)). The inclusion of J(t) improves the predictions, with an RMSE for T_max_ = 2.56, T_ave_ = 1.65 and T_min_ = 2.09 (°C), achieving better results than the classical statistical methods (typical year T_ave_ = 3.64 °C; weighted moving mean T_max_ = 2.76, T_ave_ = 1.81 and T_min_ = 2.52 (°C); linear regression T_ave_ = 1.85 °C; and Fourier T_max_ = 3.75, T_ave_ = 2.67 and T_min_ = 3.34 (°C)) for one year. The ANN models for hourly estimation have 24 neurons in the output layer (T_0h(t)_, …, T_23h(t)_) corresponding to the mean hourly temperature. In this case, the inclusion of the day of the year (J(t)) does not significantly improve the estimations, with an RMSE = 1.25 °C, but it improves the results of the ASHRAE method, which obtains an RMSE = 2.36 °C for one week. The results obtained, with lower prediction errors than those achieved with the classical methods, confirm the interest in using the ANN models for predicting temperatures in agricultural applications.

## 1. Introduction

Climate is an important factor in agricultural production, since meteorological variables are decisive in promoting plant growth and causing multiple physiological phenomena. Thus, accurate data are necessary for studying the growth and development of crops [1]. Specifically, ambient temperature is one of the key variables for crop prediction, especially in precision agriculture [2].

Early crop yield forecasts provide valuable information for growers and the industry. Thanks to the effectiveness of these forecasts, you can base your raw material transformation decisions [3]. Furthermore, the use of global climate models provides considerable information for the simulation of agricultural impacts [4].

Current crop models develop the growth process with daily temporal resolution (although some crop models require hourly diurnal temperature patterns) and use meteorological variables as inputs. On the other hand, there is the spatial resolution of the models that are developed for their application in one place and are validated based on specific data from the farm to be studied [5]. The spatial and temporal scales of numerical weather and crop simulation models are not the same. General weather models operate over very wide areas, but crop simulation models are designed to use the information at the scale of a farm. Crop simulation models run in daily interval steps and use little seasonal information [6].

The climatic variables that influence crop prediction are precipitation, temperature, solar radiation, humidity, and wind speed [7]. Ambient temperature is used to simulate crop development and growth processes (e.g., leaf expansion, photosynthesis, and respiration). The evaporation and evapotranspiration estimation models require meteorological observations as input, with data quality, which determines the quality of the estimation [8].

The time series of ambient temperature are studied to make their prediction from different modes: one of these approaches analyzes them with artificial neural networks (ANNs), which can make full use of certain unknown and hidden information in climate data, although this information cannot be extracted directly [9].

The prediction in the time series of the daily maximum, average, and minimum temperature is carried out by Ustaoglu et al. [10] in Geyve and Sakarya, Marmara region, Turkey, using three ANN models for each temperature and looking for the architecture for the best fit; from the data of the seven days prior to the prediction, the ANN configurations obtained the best RMSE settings for temperature (maximum, average, and minimum) in Geyve (3.49, 2.08, 2.47) °C and in Sakarya (3.75, 2.31, 2.48) °C. These results show that using the appropriate predictive variables achieves better results than searching for the best ANN configuration. Gos et al. [11] estimate the time series of daily maximum and minimum air temperatures over a six-year period for various climatic locations in Europe using a combination of models. Ju-Young et al. [12] propose a hybrid model for the global climate model using machine learning to generate seasonal forecasts, up to 90 days, of the average daily air temperature (RMSE, 1.02–3.35).

The prediction of the maximum and minimum temperature in four locations in the western Himalayas [13], to help forecast dangerous situations due to snow avalanches, uses an ANN model from the inputs (maximum, minimum, and dry bulb temperatures corresponding to the previous day, the cloudiness, and precipitation of the place), obtaining information for five winter seasons (June 2005 to October 2009) in a variation of the RMSE corresponding to the maximum and minimum temperature of 2.18 to 2.48 °C and 1.99 to 2.78 °C, respectively. These data indicate that making seasonal models for each time of the year can give better results than an ANN model for the whole year.

Predicting the mean daily environmental temperature is carried out with an artificial neural network (ANN) model in Denizli, southwestern Turkey [14]. The predictions are developed by taking different numbers of hidden layer neurons between 3 and 30. The best result is achieved when the number of neurons is six. The selected multilayer ANN model consists of three inputs (e.g., month of the year, day of the month, and mean temperature of the previous day) and six hidden neurons. Regarding the 2006 tests, these values are RMSE = 1.9655 and R^2^ = 0.9888.

The estimation of the mean hourly temperature using an ANN model carried out in Denizli, Turkey [15], from the time of day, the day of the month, the month of the year, and the temperature of one and two previous hours, for the simulation of the year 2005 obtains R^2^, RMSE, and MAPE values of 0.9999, 0.91301, and 0.20907%, respectively.

The article by Mihalakakou et al. [16] studies, analyzes, and models the time series of ambient air temperature on a hill in the center of Athens, Greece, using the ANN approach to extract knowledge of its past values and predict future hourly values, obtaining better predictions than the autoregressive model. Case 1-delay, using five consecutive values of the time series to predict the next value of the hourly temperature in the ambient air, obtains mean square errors (MSE) of 0.8 for January and 0.7 for July. It also studies the multi-lag case where the predicted values are added to the database at the ANN input and are used to predict future values.

In this study, we study the daily ambient temperature (maximum, average, and minimum) for the next day—i.e., (T_max_(t + 1), T_ave_(t + 1), and T_min_(t + 1))—and its estimation of the hourly mean distribution—i.e., (T_0h_(t), …, T_23h_(t))—through ANN-based models and using data measured from Mansilla Mayor, León, Castile and León, Spain. The main objective of this work was to achieve better daily and hourly predictions of ambient temperature for agricultural purposes to obtain a better prediction of (T(t + 1)) and estimation of (T_0h_(t), …, T_23h_(t)), using the least possible number of inputs. This is the main novelty of the study, to facilitate the practical application of this type of prediction model in the evaluation of irrigation needs and other agricultural processes ([17,18,19]), so as to predict plant growth or disease control [20]. For this, different ANNs were built with temperature data (maximum, average, and minimum), resulting in a better simulation fit than that achieved with the classic models, which were slightly optimized by adding the day of the year (J(t)) as a predictor, which considers the trend at the time of year.

## 2. Materials and Methods

This section describes: (1) the ambient temperature data used with the experimented models, the agrometeorological station in which they were recorded, and the place where it is located; (2) the ANN models designed for the prediction of the daily and hourly ambient temperature; and (3) the classical models (i.e., CENSOLAR typical year, weighted moving mean (WMM) with partial autocorrelation, linear regression, Fourier analysis, and ASHRAE method for hourly ambient temperature) used for their comparison.

### 2.1. Ambient Temperature Data

The ambient temperature data used in this study, during a period of eight years (i.e., 2004–2011), were collected at the agrometeorological station belonging to the Agroclimatic Information System for Irrigation (SIAR), located in Mansilla Mayor (León, Castile, and León, north-central Spain), with geographic coordinates 42°30′43″ N and 5°26′46″ W, altitude 791 m.a.s.l. and local time GMT-21.725555. SIAR is a project of the Ministry of Environment and Rural Areas and Maritime of Spain, which is managed by the Agricultural Technological Institute in Castile and León (ITACyL), through the meteorological information service InfoRiego [21], through which farmers obtain advice on irrigation water to rationalize it.

The ambient temperature in the reference agrometeorological station was measured with a Pt-1000 temperature sensor, which is based on the variation of platinum resistance with temperature. The electronic circuit for the linearization and amplification of this sensor is located next to the Vaisala HMP45C probe (Campbell Scientific, Inc., North Logan, UT, USA) to measure the ambient temperature and relative humidity, in the ranges of −40 at 60 °C, and 0 to 100%, respectively. The climatic classification of the location of the agrometeorological station is Csb [22], according to the Köppen–Geiger climatic classification, with the following annual average values (data from 1981–2010):Mean of average temperature = 11.1 °C;Maximum average temperature = 16.7 °C;Minimum average temperature = 5.5 °C;Average precipitation = 515 mm;Average number of days with precipitation equal to or greater than 1 mm = 75;Average number of clear days = 83;Annual average number of hours of sunshine = 2673 h.

### 2.2. Prediction of Ambient Temperature Using Artificial Neural Networks

The prediction of the ambient temperature (maximum, average, and minimum) of the next day ((T_max_(t + 1), T_ave_(t + 1), T_min_(t + 1)), °C), was carried out with two empirical models of the black-box type, implemented with artificial neural networks (ANNs) that use data corresponding to the seven years from 2004 to 2010. The automatic pre-processing and post-processing of the data was carried out with the Graphical User Interface (GUI) Neural Network Fitting Toolbox (*nftool*) MATLAB [23].

To achieve the proposed objective, two models with ANN architecture (i.e., ANN-1d and ANN-2d) were designed, with different combinations of input data:(T_max_(t)), the maximum daily temperature of the current day, °C;(T_ave_(t)), the average daily temperature of the current day, °C;(T_min_(t)), the daily minimum temperature of the current day, °C;(J(t)), number of the day of the year (1, …, 365), dimensionless.

In the hidden layer of the two ANN models evaluated, different numbers of neurons were used (i.e., 1–10, 20, 30, 40, and 50) to determine their influence on the precision of the simulation. The output layer in the two ANN models tested had three neurons corresponding to the predictions (T_max_(t + 1), T_ave_(t + 1), T_min_(t + 1)) for a whole year (2011), calculating the root mean square error (RMSE, °C), which was obtained in these predictions for the measured data, to select an ANN model with the best predictive behavior. The ANN architectures evaluated are shown in Figure 1a,b.

The estimation of the hourly ambient temperature during the 24 h of the same day ((T_0h_(t), T_1h_(t), …, T_23h_(t)), °C), was carried out with two empirical models of the black-box type, implemented with ANNs that use data corresponding to the seven years 2004–2010. The ANN architectures evaluated are shown in Figure 1c,d. To achieve the proposed objective, two models with ANN architecture (i.e., ANN-1h and ANN-2h) were designed, with the same combination of input data used previously in the prediction of daily temperature values.

In the hidden layer of the two ANN models evaluated, two numbers of neurons were used (i.e., 28 and 26) for the ANN-1h and ANN-2h models, respectively. The output layer in the two ANN models tested had 24 neurons corresponding to the estimates of the hourly mean ambient temperature (T_0h_(t), T_1h_(t), …, T_23h_(t)) for a whole week (Week: 10, …, 16 May 2011), calculating the root mean square error (RMSE, °C), which was obtained in these estimates for the measured data, to select an ANN model with better predictive behavior.

The ANNs are created with the *feedforwardnet* function, dimensioned with the input and output data vectors, which determine the size of the respective layers, generating a perceptron multilayer feedforward (MLP) ANN with a single hidden layer, where the function of activation selected for neurons in the hidden layer is the hyperbolic sigmoid tangent (*tansig*), while the selected transfer function for neurons in the output layer is linear (*purelin*) [24,25,26].

The Levenberg–Marquardt (BP-LM) back-propagation algorithm is applied to achieve rapid optimization (*trainlm*), as well as the following options: bias learning function and weighting moment with descending gradient (*learngdm*); normalized squared error function (*mse*); and input matrix element processing functions, such as data processing to recode unknown data rows (*fixunknowns*), repeated data vectors in the input, which do not provide useful information (*removeconstantrows*), and matrix processing to normalized vectors with minimum and maximum values in the range of −1 to 1 (*mapminmax*) [24,25,26].

ANN training was performed with the *train* function, with input and output data vector matrices over a period of seven years. (i.e., from 2004 to 2010, without considering 2011, as it is the year of verification), registering the entire training period (*epoch* and *performance* function). Finally, the *sim* function was used, with the ANNs previously trained, to make the prediction of (T_max_(t + 1), T_ave_(t + 1), T_min_(t + 1)), with the vector matrix of input data from 2011, and to make the estimation of (T_0h_(t), T_1h_(t), …, T_23h_(t)), with the vector matrix of input data from 10, …, 16 May 2011.

### 2.3. Prediction of Ambient Temperature Using Classic Models

The classical statistical models for predicting ambient air temperature, which has been considered in this work, are: (1) CENSOLAR typical year; (2) weighted moving average with partial autocorrelation; (3) linear regression; (4) Fourier analysis; and (5) the ASHRAE method for hourly ambient temperature. The construction of the models was carried out by taking the ambient temperature in the air corresponding to the seven years (2004–2010) that were considered for the construction of the models.

#### 2.3.1. CENSOLAR Typical Year

The daytime ambient temperature values, included in the CENSOLAR tables [27], characterize an average day of each month in a typical year. This is why this model is mainly applicable to large geographic areas and for long periods. The information is available for each of the Spanish provinces.

#### 2.3.2. Weighted Moving Mean (WMM) with Partial Autocorrelation

Partial autocorrelation refers to the dependence of the value of the variable with the same values of the preceding variable in time. For this reason, the weighted moving mean (WMM) model with partial autocorrelation is applied, giving more weight to the values closest to the simulation day and less weight to the furthest ones, with the aim that the average value behaves more agilely than when using a simple moving mean model.

In the present work, partial autocorrelation coefficients obtained over seven years were used, which were calculated with the MATLAB *parcorr* function, defining the weighted moving mean with time lags from 2 to 20 days, in 2011. The coefficients of partial autocorrelation are applied to the value of room temperature corresponding to its delay day, making the sum of these products and dividing by the sum of those coefficients.

#### 2.3.3. Linear Regression

The prediction of the ambient temperature can be carried out employing a linear regression, which models the relationship between a dependent variable—in this case, the ambient temperature (maximum, average, and minimum) the next day (T_max_(t + 1), T_ave_(t + 1), and T_min_(t + 1)), and an independent variable, in this case, the ambient temperature (maximum, average, and minimum) of the current day (T_max_(t), T_ave_(t), and T_min_(t)), along with a random term calculated using the MATLAB Curve Fitting Toolbox (*cftool*).

#### 2.3.4. Fourier Analysis

Fourier analysis is applied to variables that present significant frequencies [9,28], as occurs with room temperature (maximum, average, and minimum). In this case, the MATLAB Curve Fitting Toolbox (*cftool*) was used to calculate the coefficients from the 1st to the 8th harmonic.

#### 2.3.5. ASHRAE Method for Hourly Ambient Temperature

The cycle of daily solar irradiation conditions that the minimum ambient temperature usually occurs shortly before dawn, and that the maximum ambient temperature occurs between one and four hours after solar noon. Then, the daily hourly ambient temperature curve is divided into two periods: a warming period in the morning and a cooling period in the afternoon. The ASHRAE method provides a value for the ambient air temperature during each hour of the day, i.e., T_ASHRAE_ = T_max_ − (f_hour_ (T_max_ − T_min_)), as a function of the daily maximum and minimum ambient temperature, and a factor for every hour of the day in Table 1 [29].

## 3. Results and Discussion

This section includes the results obtained by the models in the prediction of the daily ambient temperature (maximum, average, and minimum) of the next day (t + 1), and the estimation of the hourly ambient temperature of the same day (t). For this purpose, measurements taken by the agrometeorological station of the SIAR network, located in Mansilla Mayor (León, Castile and León, Spain), were used. During the years 2004–2010, the data were applied for the training of the ANN models and the execution of the classic models, using the data measured during the year 2011 from the same station for the validation of the models. A comparison of the results obtained by the two types of models was made using the statistics: root mean square error (RMSE, °C), using Equation (1); coefficient of determination (R^2^), as an indicator of the level of fit of the model, utilizing Equation (2); the Durbin–Watson coefficient (DW), used to detect first-order self-correction between the data, using Equation (3); the mean percentage error (MPE), which allows the interpretation of the bias in the prediction error, employing Equation (4); forecast accuracy (FA), which is used in short-term forecasting models, using Equation (5); and the Akaike information criterion (AIC), used to select the most appropriate model taking into account the number of variables used, penalizing the more complex models, using Equation (6).
(1)$$\mathrm{RMSE}=\sqrt{\frac{\sum_{i=1}^{n}{\left(Y_{i}-{\hat{Y}}_{i}\right)}^{2}}{n}}$$
(2)$$R^{2}=1-\frac{\sum_{i=1}^{n}{\left(Y_{i}-{\hat{Y}}_{i}\right)}^{2}}{\sum_{i=1}^{n}{\left(Y_{i}-\bar{Y}\right)}^{2}}$$
(3)$$\mathrm{DW}=\frac{\sum_{i=1}^{n}{\left(\left(Y_{i}-{\hat{Y}}_{i}\right)-\left(Y_{i+1}-{\hat{Y}}_{i+1}\right)\right)}^{2}}{\sum_{i=1}^{n}{\left(Y_{i+1}-{\hat{Y}}_{i+1}\right)}^{2}}$$
(4)$$\mathrm{MPE}=\frac{\sum_{i=1}^{n}\left(\frac{\left(Y_{i}-{\hat{Y}}_{i}\right)}{Y_{i}}\right)}{n}$$
(5)$$\mathrm{FA}=\frac{\sum_{i=1}^{n}\left(1-\left|\frac{\left(Y_{i}-{\hat{Y}}_{i}\right)}{Y_{i}}\right|\right)}{n}$$
(6)$$\mathrm{AIC}=\sqrt{\frac{\sum_{i=1}^{n}{\left(Y_{i}-{\hat{Y}}_{i}\right)}^{2}}{n}}\times e^{\left(\frac{2k}{n}\right)}$$

### 3.1. Results of Predictions with Artificial Neural Network Models (ANNs)

First, the research focused on ANN models to identify the network architecture that most accurately simulates (T_max_(t + 1), T_ave_(t + 1), T_min_(t + 1)), as a function of the number of neurons (i.e., 1–10, 20, 30, 40, and 50) that make up the hidden layer. The statistic used for the validation is the RMSE of the simulated output concerning the data measured for the same variable. The results obtained for the ANN-1d model are shown in Figure 2. The best results of the prediction developed for the output variable (T_max_(t + 1)) were achieved with the architectures (3-4-3) and (3-8-3), with RMSE = 2.75 °C, while for the output variable (T_ave_(t + 1)), the best results were obtained with the architecture (3-5-3), with RMSE = 1.75 °C. Finally, the best behavior for the output variable (T_min_(t + 1)) was obtained with the architecture (3-5-3), with an RMSE = 2.11 °C. Table 2 shows the other statistics analyzed for the architecture (3-8-3). It is observed that the prediction error is much lower for the daily mean temperature than for the daily minimum temperature, and for the daily maximum temperature, the worst result is achieved.

In addition, from the results obtained with the simulation using the ANN-2d model (Figure 2), it can be seen that the best results of the prediction developed for the output variable (T_max_(t + 1)) were with the architecture (4-5-3), RMSE = 2.56 °C, while for the output variable (T_ave_(t + 1)), the best results were obtained with the architectures (4-3-3), (4-5-3), and (4-10-3), with RMSE = 1.65 °C. Finally, the best behavior for the output variable (T_min_(t + 1)) was obtained with the architecture (4-3-3), with an RMSE = 2.09 °C. Table 2 shows the other statistics analyzed for the architecture (4-8-3). It is observed that the prediction error is slightly lower when the entry of the day of the year (J(t)) is added for the three temperatures, which may have to do with the greater persistence of the daily temperature (e.g., daily solar irradiation [26]).

The estimation of the hourly mean ambient air temperature of the same day (T_0h_(t), T_1h_(t), …, T_23h_(t)), which was carried out using the ANN-1h model, from the input variables: daily maximum, average, and minimum temperature of the same day (T_max_(t), T_ave_(t), T_min_(t)), with 28 neurons in the hidden layer of the artificial neural network. As a result, in the output simulation shown in Figure 3, together with the data measured during week 10, …, 16 May 2011, it obtains an RMSE = 1.25 °C and other statistics analyzed in Table 3.

In addition, the result obtained with the simulation using the ANN-2h model is shown in Figure 3: from the input variables daily maximum, average, and minimum temperature of the same day and the day of the year (T_max_(t), T_ave_(t), T_min_(t), J(t)), with 26 neurons in the hidden layer of the artificial neural network, it obtained an RMSE = 1.42 °C and other statistics analyzed in Table 3.

The ANN-2h model, for the case of the inclusion of the day of the year (J(t)), does not provide significant improvement in the effectiveness of the estimation of the hourly distribution, as it depends on the particular climatic conditions of the specific day and not the trend at the time of year, which is what this variable contributes.

### 3.2. Results of the Classic Models

The behavior of the prediction models based on RNAs was compared with the behavior of the models that use classical techniques for predicting ambient temperature. The classic prediction models that were analyzed are CENSOLAR typical year, weighted moving mean, linear regression, Fourier analysis, and the ASHRAE method for hourly ambient temperature. In all cases, the basis of the analysis was the same, corresponding to the ambient temperature of the next day during 2011 and its comparison with the same variables measured by the agrometeorological station located in Mansilla Mayor (León, Spain), which belongs to SIAR. The figures show the results of the predictions for each day of the year. The tables summarize the analysis of fit between simulated and measured values. In the following subsubsections, the results obtained with each of the classic simulation models are analyzed.

#### 3.2.1. CENSOLAR Typical Year

The values of the daily average ambient temperature during the daytime hours of a typical year in the province of León (Spain), which are found in the CENSOLAR tables [27], are represented together with the data of the average daily temperature of the SIAR database in Mansilla Mayor (León) during 2011 in Figure 4, obtaining an RMSE = 3.64 °C and other statistics analyzed in Table 4.

The CENSOLAR typical year can be used for a first approximation because it is quick and easy to apply. The model does not take into account the daily variations for each day of the month (i.e., it offers the same data for all days of the same month), and its spatial resolution is provincial (i.e., it offers the same data for all places in the same province).

#### 3.2.2. Weighted Moving Mean (WMM) with Partial Autocorrelation

The partial autocorrelation coefficients that resulted when using the SIAR data in Mansilla Mayor (León) of the daily ambient temperature (maximum, average, and minimum) studied during seven years (2004–2010), between 1 and 20 days, are shown in Table 5. It can be observed that the dependence of the ambient temperature—(T_max_(t + 1),T_ave_(t + 1)) and (T_min_(t + 1))—on a particular day is maximum (coefficients of partial autocorrelation, 0.9326, 0.9579 and 0.8961, with the previous day, (T_max_(t),T_ave_(t)) and (T_min_(t)), respectively), decreasing the dependence very quickly with the ambient temperature received the previous days: 0.0817, −0.0863, and 0.0331 for two days (T(t − 1), T(t)), 0.1006, 0.1626, and 0.1239 for three days (T(t − 2), T(t − 1), T(t)), 0.0988, 0.1057, and 0.1303 for four days (T(t − 3), T(t − 2), T(t − 1), T(t)), until obtaining 0.0192, −0.0049, and 0.0160 for 20 days (T(t − 19), T(t − 18), …, T(t)).

Subsequently, for the prediction (T(t + 1)) during 2011, the WMM model was used with the partial autocorrelation coefficients that corresponded to the period between 2 and 20 days of delay, obtaining the simulation errors shown in Figure 5. The best results for the prediction made for the output variable (T_max_(t + 1)) were achieved with a delay of 6 days, with RMSE = 2.75 °C, while for the output variable (T_ave_(t + 1)), the best results were obtained with a delay of 5 days, with RMSE = 1.81 °C. Finally, the best result for the output variable (T_min_(t + 1)) was obtained with a delay of 10 days, with RMSE = 2.50 °C.

The model of the weighted moving mean with 2 days of delay for the daily ambient temperature (maximum, average, and minimum) is shown in Equations (7)–(9), respectively.
(7)$${\hat{T}}_{\max}\left(t+1\right)=\frac{0.9326{\;T}_{\max}\left(t\right)+0.0817{\;T}_{\max}\left(t-1\right)}{0.9326+0.0817}$$
(8)$${\hat{T}}_{\mathrm{ave}}\left(t+1\right)=\frac{0.9579{\;T}_{\mathrm{ave}}\left(t\right)-0.0863{\;T}_{\mathrm{ave}}\left(t-1\right)}{0.9579-0.0863}$$
(9)$${\hat{T}}_{\min}\left(t+1\right)=\frac{0.8961{\;T}_{\min}\left(t\right)+0.0331{\;T}_{\min}\left(t-1\right)}{0.8961+0.0331}$$

The simulation carried out for the year 2011 is represented with the WMM equation obtained in Equations (7)–(9), together with the SIAR data, which obtain an RMSE = 2.86, 1.86, and 2.52 °C, respectively, in Figure 6, and other statistics analyzed in Table 6.

The model of the weighted moving mean with 5 days of delay for the daily ambient temperature (maximum, average and minimum) is shown in Equations (10)–(12), respectively, which obtains an RMSE = 2.76, 1.81, and 2.53 °C and other statistics analyzed in Table 6.
(10)$${\hat{T}}_{\max}\left(t+1\right)=\frac{0.9326{\;T}_{\max}\left(t\right)+0.0817{\;T}_{\max}\left(t-1\right)+0.1006{\;T}_{\max}\left(t-2\right)+0.0988{\;T}_{\max}\left(t-3\right)+0.0816{\;T}_{\max}\left(t-4\right)}{0.9326+0.0817+0.1006+0.0988+0.0816}$$
(11)$${\hat{T}}_{\mathrm{ave}}\left(t+1\right)=\frac{0.9579{\;T}_{\mathrm{ave}}\left(t\right)-0.0863{\;T}_{\mathrm{ave}}\left(t-1\right)+0.1620{\;T}_{\mathrm{ave}}\left(t-2\right)+0.1057{\;T}_{\mathrm{ave}}\left(t-3\right)+0.0757{\;T}_{\mathrm{ave}}\left(t-4\right)}{0.9579-0.0863+0.1620+0.1057+0.0757}$$
(12)$${\hat{T}}_{\min}\left(t+1\right)=\frac{0.8961{\;T}_{\min}\left(t\right)+0.0331{\;T}_{\min}\left(t-1\right)+0.1239{\;T}_{\min}\left(t-2\right)+0.1303{\;T}_{\min}\left(t-3\right)+0.0994{\;T}_{\min}\left(t-4\right)}{0.8961+0.0331+0.1239+0.1303+0.0994}$$

WMM can be used for a good prediction, improving the results of the typical CENSOLAR year, making a daily prediction on the values recorded in the previous day, and using several days of delay slightly improves the predictions. Partial autocorrelation coefficients close to one confirm the persistence of the ambient temperature from one day to the next. The strongest partial autocorrelation occurs with a delay of one day, decreasing dramatically for the following days.

#### 3.2.3. Linear Regression

The prediction of the average daily temperature for tomorrow was carried out with linear regression or linear adjustment, which is a mathematical method that models the relationship between a dependent variable, in this case, the value of the average daily temperature for tomorrow (T_ave_(t + 1)), the independent variables, in this case, the value of today’s average daily temperature (T_ave_(t)), and a random term. This model can be expressed as Equation (13).
(13)$$T_{\mathrm{ave}}\left(t+1\right)=0.9576{\;T}_{\mathrm{ave}}\left(t\right)+0.4357$$
obtained from the data of the average daily temperature for the years 2004–2010 SIAR in Mansilla Mayor (León), with an RMSE = 1.96 °C, which is shown in Figure 7.

The simulation carried out for the year 2011 is represented with the linear regression equation obtained (Equation (13)) together with the SIAR data, resulting in an RMSE = 1.85 °C, in Figure 8, and other statistics analyzed in Table 7.

The linear regression obtains a precision similar to the WMM in the prediction of the average daily temperature, when also using the daily persistence of the ambient temperature.

#### 3.2.4. Fourier Analysis

The prediction of the daily ambient temperature (maximum, average, and minimum) was carried out with the calculation of the coefficients of the Fourier analysis for the data from 2004 to 2010 SIAR in Mansilla Mayor (León), obtaining the typical annual functions of Fourier shown in Table 8, Table 9 and Table 10, from the 1st to the 8th harmonic.

The simulation carried out with typical annual Fourier functions obtained for the 1st harmonic is represented, together with the SIAR data for the year 2011, which obtained an RMSE = 3.84, 2.67, and 3.37 °C, in Figure 9, and other statistics analyzed in Table 11, Table 12 and Table 13 for the daily ambient temperature (maximum, average, and minimum) of harmonics 1st–8th.

Fourier analysis can be used for a first approximation. The precision of the typical annual Fourier functions is less than the WMM, as it does not take into account daily temperature variations, while the latter does.

#### 3.2.5. ASHRAE Method for Hourly Ambient Temperature

The prediction of the hourly ambient temperature was generated with the ASHRAE method, representing the estimate made together with the SIAR data for week 10, …, 16 May 2011, which obtained an RMSE = 2.36 °C, in Figure 10, and other statistics analyzed in Table 14.

This hourly distribution of temperature that rises during the morning until a few hours after solar noon, and continues descending until dawn, is typical on clear days, when the variability of the atmospheric components is not very pronounced [30]. When cloudiness appears, the intensity of solar radiation, received by the surface of the ground, decreases, then the effect of a more constant temperature is usually produced throughout those hours, and when there are sudden storms or rains, the temperature drops more suddenly.

## 4. Conclusions

ANNs can learn, as the process varies over time, which makes them interesting for the creation of models applied in the prediction of the daily ambient temperature (T_max_(t + 1), T_ave_(t + 1), T_min_(t + 1)) and the estimation of its hourly distribution (T_0h_(t), …, T_23h_(t)).

In this study, ANN inputs are established as the (T_max_(t), T_ave_(t), T_min_(t)) of the day on which we make the prediction and adding the (J(t)), the ANN output being the future value of the next day, predicted (T_max_(t + 1), T_ave_(t + 1), T_min_(t + 1)) for the daily case, and its distribution on the same day estimated (T_0h_(t), …, T_23h_(t)) for the hourly case.

The prediction of the ambient temperature of the next day (T(t + 1)) is of great interest for all types of agricultural applications, especially for estimating irrigation needs using crop evapotranspiration models (FAO Penman-Monteith, Priestley-Taylor, or Hargreaves, among others), and to predict the growth of plants or the development of diseases, as well as its application to precision agriculture. In this study, a prediction of (T_max_(t + 1), T_ave_(t + 1), T_min_(t + 1)) and an estimate of (T_0h_(t), …, T_23h_(t)) were made using ANNs, trying to design the networks with the simplest architecture and the least possible number of inputs, i.e., (T_max_(t), T_ave_(t), T_min_(t)) and (J(t)), to facilitate its practical technological application. The only requirement is to have an important and reliable set of ambient temperature data, which can be measured on site, for training the ANN models.

In the predictions made with the ANNs, current temperature data were used, obtaining models with which the predictive performance is improved compared to the classic models considered (typical CENSOLAR year, WMM with a delay of several days, linear regression, and Fourier analysis). Except for the WMM model (T_max_(t + 1)) with a 6-day delay, for which a similar fit was obtained (i.e., RMSE = 2.76 °C), but which is improved with the ANN-2d model for the architecture (4-5-3) that includes the predictor variable of the (J(t)) (i.e., RMSE = 2.56 °C). Therefore, the daily predictive improvement of the ANN models is achieved when we apply the day of the year as the input variable, thus the prediction refers to the time of year in which it occurs, and not so in the case of the hourly estimate by depending on its distribution of the particular climatic conditions of the concrete day.

To continue working on the predictive capacity and applicability of ambient temperature, the following lines of research are proposed:

(1) The use of other ANN input variables (e.g., humidity, atmospheric pressure, cloudiness) that help detect changes in the evolution of temperature, mainly days with sudden changes in the weather.

(2) The creation of ANN models for different times/seasons of the year with similar climatological characteristics.

(3) The use of forecasts from national meteorological services as input data for the ANN model, in addition to the historical data recorded in the area.

## Figures and Tables

**Figure 1 sensors-22-04850-f001:**
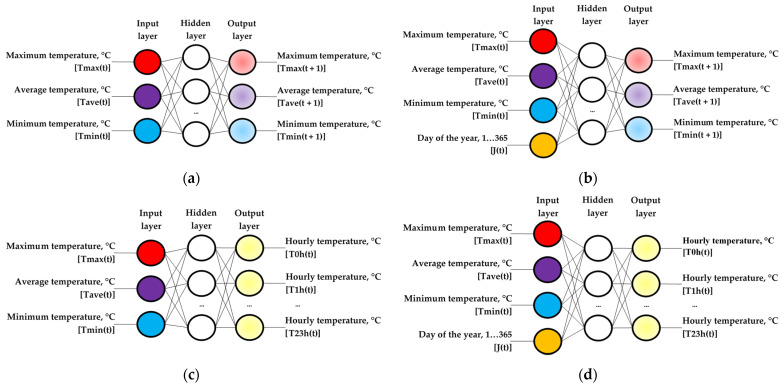
Architecture of the models with artificial neural networks (ANNs) evaluated for daily temperature, based on the input variables: (**a**) ANN-1d model, with three inputs (T_max_(t), T_ave_(t), T_min_(t)); (**b**) ANN-2d model, with four inputs (T_max_(t), T_ave_(t), T_min_(t), J(t)); and for hourly temperature, based on the input variables: (**c**) ANN-1h model, with three inputs (T_max_(t), T_ave_(t), T_min_(t)); (**d**) ANN-2h model, with four inputs (T_max_(t), T_ave_(t), T_min_(t), J(t)).

**Figure 2 sensors-22-04850-f002:**
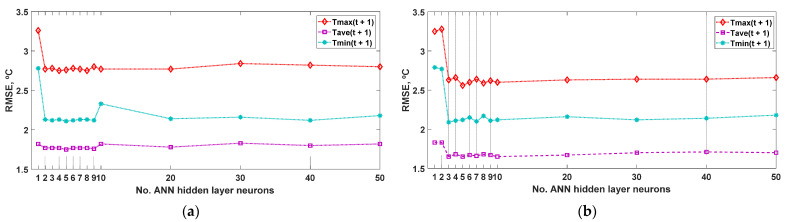
Effectiveness of the prediction of the daily ambient temperature the next day (maximum, T_max_; average, T_ave_; and minimum, T_min_) simulated with the ANN model, compared to the values of the measured variable, in 2011, as a function of the number of neurons in the hidden layer: (**a**) ANN-1d model, with three inputs (T_max_(t), T_ave_(t), T_min_(t)); (**b**) ANN-2d model, with four inputs (T_max_(t), T_ave_(t), T_min_(t), J(t)). RMSE: root mean square error (°C).

**Figure 3 sensors-22-04850-f003:**
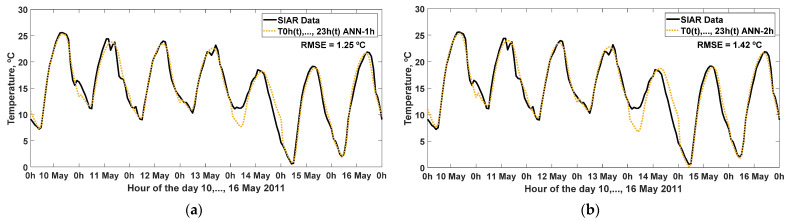
Effectiveness of the estimation of hourly ambient temperature on the same day (hour 0, T_0h_; hour 1, T_1h_; …; hour 23, T_23h_) simulated with the ANN model, compared to the measured values of the variable, at 10, …, 16 May 2011: (**a**) ANN-1h model, with three inputs (T_max_(t), T_ave_(t), T_min_(t)); (**b**) ANN-2h model, with four inputs (T_max_(t), T_ave_(t), T_min_(t), J(t)). RMSE: root mean square error (°C).

**Figure 4 sensors-22-04850-f004:**
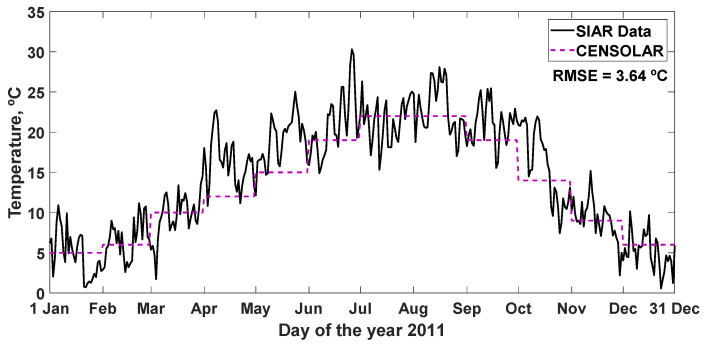
Recommended values of the average daily ambient temperature during the hours of daytime, in a typical year CENSOLAR (2009), in the province of León (Spain), and the data measured in Mansilla Mayor (León) by the SIAR network during 2011. RMSE: root mean square error (°C).

**Figure 5 sensors-22-04850-f005:**
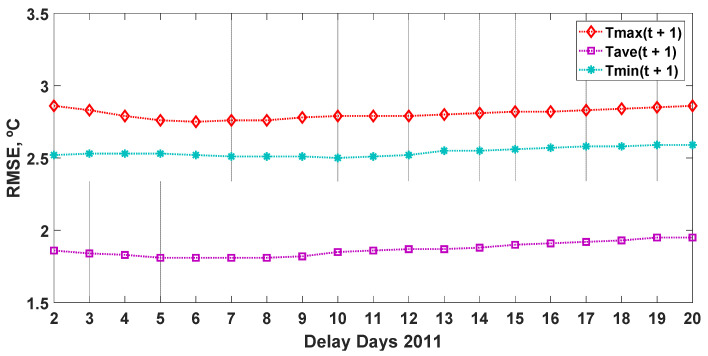
Effectiveness of the prediction of the daily ambient temperature (maximum, T_max_; average, T_ave_; and minimum, T_min_) on the following day, simulated with the weighted moving mean (WMM) model, together with the values of the variable measured in 2011, depending on the number of delays considered (from 2 to 20 days). RMSE: root mean square error (°C).

**Figure 6 sensors-22-04850-f006:**
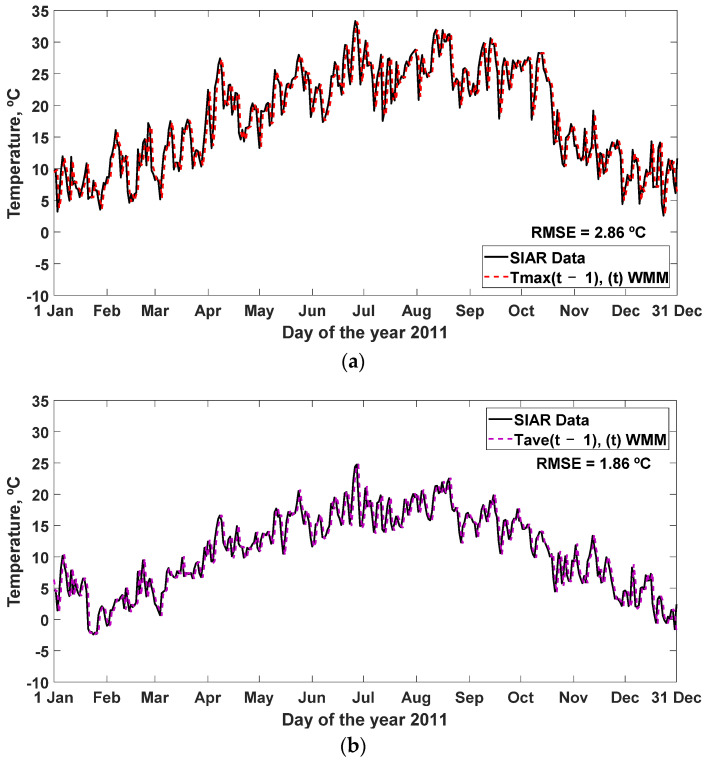
Effectiveness of the prediction of the daily ambient temperature for the next day, simulated with the weighted moving mean (WMM) model, together with the measured values of the variable in 2011, as a function of the number of lags considered (2 days): (**a**) WMM model, for output (T_max_(t)); (**b**) WMM model, for output (T_ave_(t)); (**c**) WMM model, for output (T_min_(t)). RMSE: root mean square error (°C).

**Figure 7 sensors-22-04850-f007:**
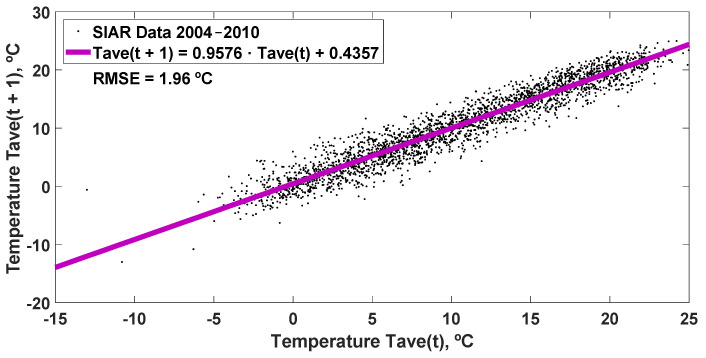
Equation of the linear regression with a delay of one day, resulting from the data of the average daily temperature for the years 2004–2010 SIAR in Mansilla Mayor (León). RMSE: root mean square error (°C).

**Figure 8 sensors-22-04850-f008:**
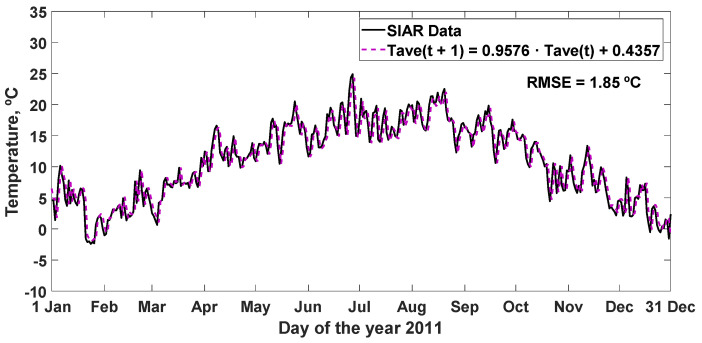
Effectiveness of the prediction of the average daily temperature for the next day, simulated with the linear regression, together with the values of the variable measured in 2011. RMSE: root mean square error (°C).

**Figure 9 sensors-22-04850-f009:**
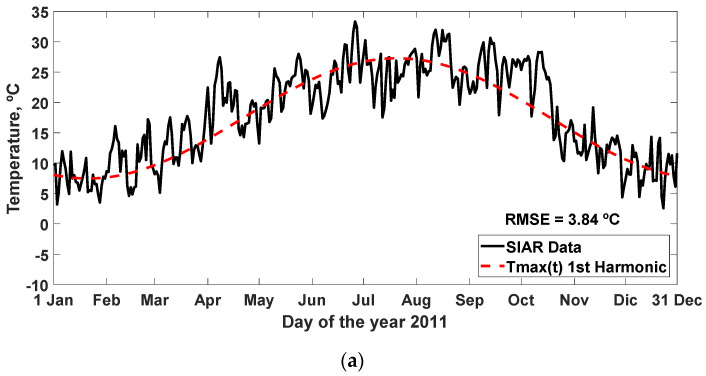
Data of the daily temperature (maximum, average, and minimum) of the year 2011 SIAR in Mansilla Mayor (León) and typical annual Fourier function for the 1st harmonic: (**a**) Fourier model for output (T_max_(t)); (**b**) Fourier model for output (T_ave_(t)); (**c**) Fourier model for output (T_min_(t)). RMSE: root mean square error (°C).

**Figure 10 sensors-22-04850-f010:**
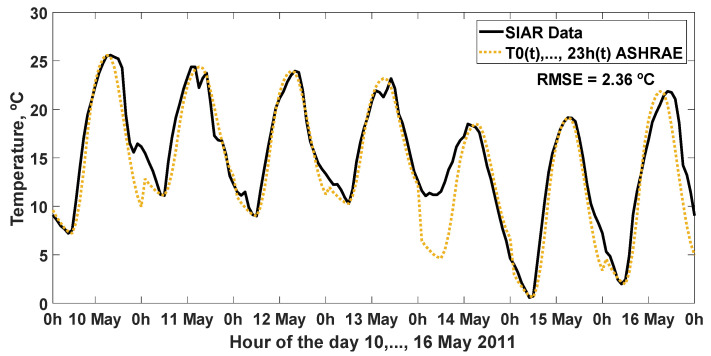
SIAR hourly average temperature data in Mansilla Mayor (León) and value resulting from the estimation made with the ASHRAE method for week 10, …, 16 May 2011. RMSE: root mean square error (°C).

**Table 1 sensors-22-04850-t001:** ASHRAE hourly factor of ambient air temperature (f_hour_).

ASHRAE Hourly Factor (f_hour_): Ambient Air Temperature																							
0 h	1 h	2 h	3 h	4 h	5 h	6 h	7 h	8 h	9 h	10 h	11 h	12 h	13 h	14 h	15 h	16 h	17 h	18 h	19 h	20 h	21 h	22 h	23 h
0.87	0.91	0.94	0.97	0.99	1.00	0.94	0.82	0.65	0.44	0.27	0.15	0.07	0.02	0.00	0.01	0.07	0.18	0.31	0.45	0.58	0.70	0.79	0.85

**Table 2 sensors-22-04850-t002:** Effectiveness of the neural prediction of daily temperature using the ANN-1d (3-8-3) and ANN-2d (4-8-3) models, with respect to the SIAR data in Mansilla Mayor (León) for the year 2011.

ANN Prediction Models: Daily Temperature (Maximum, Average, and Minimum)							
ANN	Outputs	RMSE	R^2^	DW	MPE	FA	AIC
ANN-1d (3-8-3)	T_max_(t + 1)	2.7537	0.8714	1.9245	−0.0390	0.8410	2.7991
	T_ave_(t + 1)	1.7741	0.9181	1.8461	−0.0170	0.7843	1.8021
	T_min_(t + 1)	2.1261	0.8322	1.6176	0.1010	0.6507	2.1578
ANN-2d (4-8-3)	T_max_(t + 1)	2.5886	0.8863	1.7957	−0.0256	0.8505	2.6456
	T_ave_(t + 1)	1.6756	0.9269	1.7168	0.0045	0.7772	1.7126
	T_min_(t + 1)	2.1730	0.8393	1.6303	0.0390	0.6711	2.2180

The best results of the variables (T_max_(t + 1), T_ave_(t + 1), T_min_(t + 1)) with each statistic (RMSE (°C), R^2^, DW, MPE, FA and AIC) are underlined.

**Table 3 sensors-22-04850-t003:** Effectiveness of the neural prediction of hourly temperature using the ANN-1h (3-28-3) and ANN-2h (4-26-3) models, with respect to the SIAR data in Mansilla Mayor (León) for week 10, …, 16 May 2011.

ANN Estimation Models: Hourly Mean Temperature							
ANN	Outputs	RMSE	R^2^	DW	MPE	FA	AIC
ANN-1h	T_0h_(t), …, T_23h_(t)	1.2524	0.9565	0.4339	−0.0131	0.9199	0.8766
ANN-2h	T_0h_(t), …, T_23h_(t)	1.4173	0.9444	0.3695	−0.0008	0.9081	1.0030

The best results of the variables (T_0h_(t), …, T_23h_(t)) with each statistic (RMSE (°C), R^2^, DW, MPE, FA and AIC) are underlined.

**Table 4 sensors-22-04850-t004:** Effectiveness of the CENSOLAR daytime ambient temperature tables, with respect to the SIAR data in Mansilla Mayor (León) for the year 2011.

CENSOLAR Typical Year: Average Daily Ambient Temperature during Daytime							
	Outputs	RMSE	R^2^	DW	MPE	FA	AIC
CENSOLAR	T_ave_(t)	3.6352	0.7505	0.4255	−0.1157	0.6524	3.6546

**Table 5 sensors-22-04850-t005:** Partial autocorrelation coefficients of the weighted moving mean (WMM) model, for lags of 1–20 days with the SIAR data of seven years (2004–2010) in Mansilla Mayor (León) of daily ambient temperature (maximum, average, and minimum).

Partial Autocorrelation Coefficients: Daily Temperature (Maximum, Average, and Minimum)			
Days of Delay	Maximum	Average	Minimum
1	0.9326	0.9579	0.8961
2	0.0817	−0.0863	0.0331
3	0.1006	0.1626	0.1239
4	0.0988	0.1057	0.1303
5	0.0816	0.0757	0.0994
6	0.0734	0.0855	0.0904
7	0.1014	0.0679	0.0357
8	0.0248	0.0105	0.0222
9	0.0600	0.0468	0.0459
10	0.0490	0.0689	0.0593
11	0.0462	0.0332	0.0598
12	0.0255	0.0568	0.0397
13	0.0384	0.0024	0.0637
14	0.0448	0.0254	0.0170
15	0.0225	0.0545	0.0354
16	0.0249	0.0204	0.0286
17	0.0085	0.0289	0.0578
18	0.0540	0.0320	0.0246
19	0.0406	0.0454	0.0257
20	0.0192	−0.0049	0.0160

The best results of the variables (T_max_(t), T_ave_(t), T_min_(t)) with each statistic (RMSE (°C), R^2^, DW, MPE, FA and AIC) are underlined.

**Table 6 sensors-22-04850-t006:** Effectiveness of the prediction of the weighted moving mean (WMM) using the partial autocorrelation coefficients for 2 and 5 days of temperature lag (maximum, average, and minimum), with respect to the SIAR data in Mansilla Mayor (León) for the year 2011.

WMM Prediction Models: Daily Temperature (Maximum, Average, and Minimum)							
Inputs	Outputs	RMSE	R^2^	DW	MPE	FA	AIC
T_max_(t − 1), (t)	T_max_(t + 1)	2.8626	0.8610	1.8969	−0.0316	0.8366	2.8941
T_ave_(t − 1), (t)	T_ave_(t + 1)	1.8628	0.9097	1.9296	−0.0463	0.7534	1.7811
T_min_(t − 1), (t)	T_min_(t + 1)	2.5226	0.7740	1.8751	0.1455	0.7016	2.5458
T_max_(t – 4), …, (t)	T_max_(t + 1)	2.7562	0.8711	1.6610	−0.0380	0.8391	2.8324
T_ave_(t − 4), …, (t)	T_ave_(t + 1)	1.8135	0.9144	1.5192	−0.0116	0.7967	1.8636
T_min_(t – 4), …, (t)	T_min_(t + 1)	2.5287	0.7729	1.4721	0.2502	0.7670	2.5970

The best results of the variables (T_max_(t + 1), T_ave_(t + 1), T_min_(t + 1)) with each statistic (RMSE (°C), R^2^, DW, MPE, FA and AIC) are underlined.

**Table 7 sensors-22-04850-t007:** Effectiveness of the linear regression prediction in the average daily temperature, with respect to the SIAR data in Mansilla Mayor (León) for the year 2011.

Linear Regression Model (Equation 13): Average Daily Temperature							
Inputs	Outputs	RMSE	R^2^	DW	MPE	FA	AIC
T_ave_(t)	T_ave_(t + 1)	1.8505	0.9109	1.7523	−0.0685	0.7552	1.8580

**Table 8 sensors-22-04850-t008:** Typical annual Fourier functions of the 2004–2010 SIAR daily maximum temperature data in Mansilla Mayor (León), from the 1st to the 8th harmonic.

Harmonics	Typical Annual Fourier Function: Daily Maximum Temperature
1	$T_{\max}\left(t\right)=17.38-9.237\cos\left(0.01742\;t\right)-3.553\mathrm{sen}\left(0.01742\;t\right)$
2	$\begin{array}{l} T_{\max}\left(t\right)=16.34-10.4\cos\left(0.01574\;t\right)-0.112\mathrm{sen}\left(0.01574\;t\right)+\\ 0.7262\cos\left(2\cdot 0.01574\;t\right)+1.319\mathrm{sen}\left(2\cdot 0.01574\;t\right) \end{array}$
3	$\begin{array}{c} T_{\max}\left(t\right)=17.73-8.947\cos\left(0.01766\;t\right)-3.991\mathrm{sen}\left(0.01766\;t\right)-0.8181\cos\left(2\cdot 0.01766\;t\right)+\\ 0.9866\mathrm{sen}\left(2\cdot 0.01766\;t\right)-0.5532\cos\left(3\cdot 0.01766\;t\right)+0.1289\mathrm{sen}\left(3\cdot 0.01766\;t\right) \end{array}$
4	$\begin{array}{c} T_{\max}\left(t\right)=17.47-9.061\cos\left(0.01756\;t\right)-3.808\mathrm{sen}\left(0.01756\;t\right)-0.757\cos\left(2\cdot 0.01756\;t\right)+\\ 1.043\mathrm{sen}\left(2\cdot 0.01756\;t\right)-0.5232\cos\left(3\cdot 0.01756\;t\right)+0.1571\mathrm{sen}\left(3\cdot 0.01756\;t\right)+\\ 0.03129\cos\left(4\cdot 0.01756\;t\right)-0.09783\mathrm{sen}\left(4\cdot 0.01756\;t\right) \end{array}$
5	$\begin{array}{c} T_{\max}\left(t\right)=15.76-10.03\cos\left(0.01446\;t\right)+2.507\mathrm{sen}\left(0.01446\;t\right)+1.986\cos\left(2\cdot 0.01446\;t\right)+\\ 0.08989\mathrm{sen}\left(2\cdot 0.01446\;t\right)-0.1587\cos\left(3\cdot 0.01446\;t\right)-0.4054\mathrm{sen}\left(3\cdot 0.01446\;t\right)-\\ 0.5053\cos\left(4\cdot 0.01446\;t\right)-0.3611\mathrm{sen}\left(4\cdot 0.01446\;t\right)-0.07432\cos\left(5\cdot 0.01446\;t\right)-0.1128\mathrm{sen}\left(5\cdot 0.01446\;t\right) \end{array}$
6	$\begin{array}{c} T_{\max}\left(t\right)=17.84-8.27\cos\left(0.01822\;t\right)-4.927\mathrm{sen}\left(0.01822\;t\right)-1.08\cos\left(2\cdot 0.01822\;t\right)+\\ 0.6339\mathrm{sen}\left(2\cdot 0.01822\;t\right)-0.6738\cos\left(3\cdot 0.01822\;t\right)-0.07937\mathrm{sen}\left(3\cdot 0.01822\;t\right)-\\ 0.04703\cos\left(4\cdot 0.01822\;t\right)-0.1428\mathrm{sen}\left(4\cdot 0.01822\;t\right)-0.1463\cos\left(5\cdot 0.01822\;t\right)+\\ 0.02918\mathrm{sen}\left(5\cdot 0.01822\;t\right)-0.1591\cos\left(6\cdot 0.01822\;t\right)+0.1942\mathrm{sen}\left(6\cdot 0.01822\;t\right) \end{array}$
7	$\begin{array}{c} T_{\max}\left(t\right)=401600-241900\cos\left(0.006293\;t\right)-680000\mathrm{sen}\left(0.006293\;t\right)-404100\cos\left(2\cdot 0.006293\;t\right)+\\ 329300\mathrm{sen}\left(2\cdot 0.006293\;t\right)+255000\cos\left(3\cdot 0.006293\;t\right)+154500\mathrm{sen}\left(3\cdot 0.006293\;t\right)+\\ 26280\cos\left(4\cdot 0.006293\;t\right)-128400\mathrm{sen}\left(4\cdot 0.006293\;t\right)-41560\cos\left(5\cdot 0.006293\;t\right)+\\ 5917\mathrm{sen}\left(5\cdot 0.006293\;t\right)+4092\cos\left(6\cdot 0.006293\;t\right)+7779\mathrm{sen}\left(6\cdot 0.006293\;t\right)+\\ 618.7\cos\left(7\cdot 0.006293\;t\right)-669.5\mathrm{sen}\left(7\cdot 0.006293\;t\right) \end{array}$
8	$\begin{array}{c} T_{\max}\left(t\right)=17.26-9.431\cos\left(0.01721\;t\right)-3.149\mathrm{sen}\left(0.01721\;t\right)-0.5215\cos\left(2\cdot 0.01721\;t\right)+\\ 1.209\mathrm{sen}\left(2\cdot 0.01721\;t\right)-0.4128\cos\left(3\cdot 0.01721\;t\right)+0.2276\mathrm{sen}\left(3\cdot 0.01721\;t\right)+\\ 0.0707\cos\left(4\cdot 0.01721\;t\right)-0.1249\mathrm{sen}\left(4\cdot 0.01721\;t\right)+0.06714\cos\left(5\cdot 0.01721\;t\right)+0.08861\mathrm{sen}\left(5\cdot 0.01721\;t\right)+\\ 0.1252\cos\left(6\cdot 0.01721\;t\right)+0.03636\mathrm{sen}\left(6\cdot 0.01721\;t\right)-0.07931\cos\left(7\cdot 0.01721\;t\right)-\\ 0.2324\mathrm{sen}\left(7\cdot 0.01721\;t\right)-0.2771\cos\left(8\cdot 0.01721\;t\right)-0.3641\mathrm{sen}\left(8\cdot 0.01721\;t\right) \end{array}$

**Table 9 sensors-22-04850-t009:** Typical annual Fourier functions of the 2004–2010 SIAR daily average temperature data in Mansilla Mayor (León), from the 1st to the 8th harmonic.

Harmonics	Typical Annual Fourier Function: Daily Average Temperature
1	$T_{\mathrm{ave}}\left(t\right)=10.59-7.77\cos\left(0.01791\;t\right)-3.511\mathrm{sen}\left(0.01791\;t\right)$
2	$\begin{array}{c} T_{\mathrm{ave}}\left(t\right)=7.033-4.591\;\cos\left(0.01111\;t\right)+7.348\;\mathrm{sen}\left(0.01111\;t\right)-\\ 0.157\;\cos\left(2\cdot 0.01111\;t\right)-3.559\;\mathrm{sen}\left(2\cdot 0.01111\;t\right) \end{array}$
3	$\begin{array}{c} T_{\mathrm{ave}}\left(t\right)=10.83-7.264\cos\left(0.01837\;t\right)-4.176\mathrm{sen}\left(0.01837\;t\right)-0.4432\cos\left(2\cdot 0.01837\;t\right)+\\ 0.1391\mathrm{sen}\left(2\cdot 0.01837\;t\right)-0.5165\cos\left(3\cdot 0.01837\;t\right)-0.1922\mathrm{sen}\left(3\cdot 0.01837\;t\right) \end{array}$
4	$\begin{array}{c} T_{\mathrm{ave}}\left(t\right)=9.063-4.886\cos\left(0.01227\;t\right)+4.095\mathrm{sen}\left(0.01227\;t\right)+0.7542\cos\left(2\cdot 0.01227\;t\right)-\\ 4.555\mathrm{sen}\left(2\cdot 0.01227\;t\right)-1.292\cos\left(3\cdot 0.01227\;t\right)-0.7837\mathrm{sen}\left(3\cdot 0.01227\;t\right)-\\ 0.7717\cos\left(4\cdot 0.01227\;t\right)+0.3988\mathrm{sen}\left(4\cdot 0.01227\;t\right) \end{array}$
5	$\begin{array}{c} T_{\mathrm{ave}}\left(t\right)=9.727-8.823\cos\left(0.0161\;t\right)-0.4606\mathrm{sen}\left(0.0161\;t\right)+0.6361\cos\left(2\cdot 0.0161\;t\right)+\\ 0.6803\mathrm{sen}\left(2\cdot 0.0161\;t\right)+0.0942\cos\left(3\cdot 0.0161\;t\right)+0.3764\mathrm{sen}\left(3\cdot 0.0161\;t\right)+0.2968\cos\left(4\cdot 0.0161\;t\right)-\\ 0.09827\mathrm{sen}\left(4\cdot 0.0161\;t\right)+0.3474\cos\left(5\cdot 0.0161\;t\right)+0.06249\mathrm{sen}\left(5\cdot 0.0161\;t\right) \end{array}$
6	$\begin{array}{c} T_{\mathrm{ave}}\left(t\right)=9.946-8.676\cos\left(0.01654\;t\right)-1.251\mathrm{sen}\left(0.01654\;t\right)+0.3369\cos\left(2\cdot 0.01654\;t\right)+\\ 0.7051\mathrm{sen}\left(2\cdot 0.01654\;t\right)-0.08379\cos\left(3\cdot 0.01654\;t\right)+0.3957\mathrm{sen}\left(3\cdot 0.01654\;t\right)+\\ 0.2522\cos\left(4\cdot 0.01654\;t\right)+0.01242\mathrm{sen}\left(4\cdot 0.01654\;t\right)+0.3075\cos\left(5\cdot 0.01654\;t\right)+\\ 0.2099\mathrm{sen}\left(5\cdot 0.01654\;t\right)+0.1048\cos\left(6\cdot 0.01654\;t\right)+0.007583\mathrm{sen}\left(6\cdot 0.01654\;t\right) \end{array}$
7	$\begin{array}{c} T_{\mathrm{ave}}\left(t\right)=9.874-8.751\cos\left(0.01643\;t\right)-1.041\mathrm{sen}\left(0.01643\;t\right)+0.3864\cos\left(2\cdot 0.01643\;t\right)+\\ 0.714\mathrm{sen}\left(2\cdot 0.01643\;t\right)-0.06148\cos\left(3\cdot 0.01643\;t\right)+0.4084\mathrm{sen}\left(3\cdot 0.01643\;t\right)+\\ 0.246\cos\left(4\cdot 0.01643\;t\right)+0.00087\mathrm{sen}\left(4\cdot 0.01643\;t\right)+0.3057\cos\left(5\cdot 0.01643\;t\right)+\\ 0.1888\mathrm{sen}\left(5\cdot 0.01643\;t\right)+0.0705\cos\left(6\cdot 0.01643\;t\right)+0.009187\mathrm{sen}\left(6\cdot 0.01643\;t\right)-\\ 0.1137\cos\left(7\cdot 0.01643\;t\right)+0.1537\mathrm{sen}\left(7\cdot 0.01643\;t\right) \end{array}$
8	$\begin{array}{c} T_{\mathrm{ave}}\left(t\right)=10.47-7.99\cos\left(0.0176\;t\right)-3.034\mathrm{sen}\left(0.0176\;t\right)-0.224\cos\left(2\cdot 0.0176\;t\right)+\\ 0.4723\mathrm{sen}\left(2\cdot 0.0176\;t\right)-0.4428\cos\left(3\cdot 0.0176\;t\right)+0.1546\mathrm{sen}\left(3\cdot 0.0176\;t\right)+\\ 0.04217\cos\left(4\cdot 0.0176\;t\right)+0.04157\mathrm{sen}\left(4\cdot 0.0176\;t\right)-0.03351\cos\left(5\cdot 0.0176\;t\right)+0.3185\mathrm{sen}\left(5\cdot 0.0176\;t\right)+\\ 0.06229\cos\left(6\cdot 0.0176\;t\right)+0.191\mathrm{sen}\left(6\cdot 0.0176\;t\right)-0.1071\cos\left(7\cdot 0.0176\;t\right)-\\ 0.02178\mathrm{sen}\left(7\cdot 0.0176\;t\right)-0.2866\cos\left(8\cdot 0.0176\;t\right)-0.1989\mathrm{sen}\left(8\cdot 0.0176\;t\right) \end{array}$

**Table 10 sensors-22-04850-t010:** Typical annual Fourier functions of the 2004–2010 SIAR daily minimum temperature data in Mansilla Mayor (León), from the 1st to the 8th harmonic.

Harmonics	Typical Annual Fourier Function: Daily Minimum Temperature
1	$T_{\min}\left(t\right)=4.232-5.568\cos\left(0.01853\;t\right)-3.642\mathrm{sen}\left(0.01853\;t\right)$
2	$\begin{array}{c} T_{\min}\left(t\right)=34.26-13.72\cos\left(0.006226\;t\right)-47.1\mathrm{sen}\left(0.006226\;t\right)-\\ 21.07\cos\left(2\cdot 0.006226\;t\right)+14.31\mathrm{sen}\left(2\cdot 0.006226\;t\right) \end{array}$
3	$\begin{array}{c} T_{\min}\left(t\right)=1.642-4.772\cdot \cos\left(0.01208\cdot t\right)+5.076\cdot \mathrm{sen}\left(0.01208\cdot t\right)+1.262\cdot \cos\left(2\cdot 0.01208\cdot t\right)-\\ 2.271\cdot \mathrm{sen}\left(2\cdot 0.01208\cdot t\right)+0.5186\cdot \cos\left(3\cdot 0.01208\cdot t\right)-0.6906\cdot \mathrm{sen}\left(3\cdot 0.01208\cdot t\right) \end{array}$
4	$\begin{array}{c} T_{\min}\left(t\right)=12810000000-2.031\cos\left(0.0008166\;t\right)+2.886\mathrm{sen}\left(0.0008166\;t\right)+\\ 9.878\cos\left(2\cdot 0.0008166\;t\right)-2.865\mathrm{sen}\left(2\cdot 0.0008166\;t\right)-2.691\cos\left(3\cdot 0.0008166\;t\right)+\\ 1.213\mathrm{sen}\left(3\cdot 0.0008166\;t\right)+3.137\cos\left(4\cdot 0.0008166\;t\right)-1.986\mathrm{sen}\left(4\cdot 0.0008166\;t\right) \end{array}$
5	$\begin{array}{c} T_{\min}\left(t\right)=3.31-6.84\cos\left(0.01585\;t\right)-0.2011\mathrm{sen}\left(0.01585\;t\right)+0.8658\cos\left(2\cdot 0.01585\;t\right)+\\ 0.1112\mathrm{sen}\left(2\cdot 0.01585\;t\right)+0.3911\cos\left(3\cdot 0.01585\;t\right)+0.5454\mathrm{sen}\left(3\cdot 0.01585\;t\right)+0.4313\\ \cos\left(4\cdot 0.01585\;t\right)-0.03046\mathrm{sen}\left(4\cdot 0.01585\;t\right)+0.4672\cos\left(5\cdot 0.01585\;t\right)+0.02904\mathrm{sen}\left(5\cdot 0.01585\;t\right) \end{array}$
6	$\begin{array}{c} T_{\min}\left(t\right)=3.156-6.85\cos\left(0.01544\;t\right)+0.3841\mathrm{sen}\left(0.01544\;t\right)+1.047\cos\left(2\cdot 0.01544\;t\right)+\\ 0.01405\mathrm{sen}\left(2\cdot 0.01544\;t\right)+0.563\cos\left(3\cdot 0.01544\;t\right)+0.4533\mathrm{sen}\left(3\cdot 0.01544\;t\right)+\\ 0.4101\cos\left(4\cdot 0.01544\;t\right)-0.165\mathrm{sen}\left(4\cdot 0.01544\;t\right)+0.3898\cos\left(5\cdot 0.01544\;t\right)-\\ 0.1011\mathrm{sen}\left(5\cdot 0.01544\;t\right)-0.06743\cos\left(6\cdot 0.01544\;t\right)+0.0967\mathrm{sen}\left(6\cdot 0.01544\;t\right) \end{array}$
7	$\begin{array}{c} T_{\min}\left(t\right)=3.35-6.901\cos\left(0.01608\;t\right)-0.5126\mathrm{sen}\left(0.01608\;t\right)+0.6787\cos\left(2\cdot 0.01608\;t\right)+\\ 0.181\mathrm{sen}\left(2\cdot 0.01608\;t\right)+0.21\cos\left(3\cdot 0.01608\;t\right)+0.6252\mathrm{sen}\left(3\cdot 0.01608\;t\right)+0.3578\cos\left(4\cdot 0.01608\;t\right)+\\ 0.1079\mathrm{sen}\left(4\cdot 0.01608\;t\right)+0.4412\cos\left(5\cdot 0.01608\;t\right)+0.1948\mathrm{sen}\left(5\cdot 0.01608\;t\right)-0.02654\cos\\ \left(6\cdot 0.01608\;t\right)+0.02654\mathrm{sen}\left(6\cdot 0.01608\;t\right)+0.1169\cos\left(7\cdot 0.01608\;t\right)+0.2493\mathrm{sen}\left(7\cdot 0.01608\;t\right) \end{array}$
8	$\begin{array}{c} T_{\min}\left(t\right)=3.573-6.732\cos\left(0.01666\;t\right)-1.332\mathrm{sen}\left(0.01666\;t\right)+0.4493\cos\left(2\cdot 0.01666\;t\right)+\\ 0.1722\mathrm{sen}\left(2\cdot 0.01666\;t\right)-0.06693\cos\left(3\cdot 0.01666\;t\right)+0.574\mathrm{sen}\left(3\cdot 0.01666\;t\right)+0.2302\\ \cos\left(4\cdot 0.01666\;t\right)+0.2102\mathrm{sen}\left(4\cdot 0.01666\;t\right)+0.3272\cos\left(5\cdot 0.01666\;t\right)+0.4028\mathrm{sen}\left(5\cdot 0.01666\;t\right)+\\ 0.01972\cos\left(6\cdot 0.01666\;t\right)+0.06246\mathrm{sen}\left(6\cdot 0.01666\;t\right)-0.07979\cos\left(7\cdot 0.01666\;t\right)+0.2817\\ \mathrm{sen}\left(7\cdot 0.01666\;t\right)+0.05128\cos\left(8\cdot 0.01666\;t\right)+0.3089\mathrm{sen}\left(8\cdot 0.01666\;t\right) \end{array}$

**Table 11 sensors-22-04850-t011:** Effectiveness of the typical annual Fourier function from the 1st to the 8th harmonic of maximum temperature with respect to the SIAR data in Mansilla Mayor (León) for the year 2011.

Fourier Analysis: Daily Maximum Temperature							
Harmonics	Outputs	RMSE	R^2^	DW	MPE	FA	AIC
1	T_max_(t)	3.8353	0.7505	0.5492	−0.0106	0.7969	3.8555
2	T_max_(t)	3.7535	0.7610	0.5741	−0.0076	0.7972	3.7916
3	T_max_(t)	3.7463	0.7620	0.5760	−0.0072	0.7974	3.8061
4	T_max_(t)	3.7482	0.7617	0.5755	−0.0072	0.7975	3.8289
5	T_max_(t)	3.7520	0.7612	0.5744	−0.0110	0.7969	3.8535
6	T_max_(t)	3.7904	0.7563	0.5629	−0.0076	0.7959	3.9152
7	T_max_(t)	3.7583	0.7604	0.5717	−0.0075	0.7956	3.9042
8	T_max_(t)	3.8490	0.7487	0.5467	−0.0082	0.7925	4.0183

The best results of the variable (T_max_(t)) with each statistic (RMSE (°C), R^2^, DW, MPE, FA and AIC) are underlined.

**Table 12 sensors-22-04850-t012:** Effectiveness of the typical annual Fourier function from the 1st to the 8th harmonic of average temperature with respect to the SIAR data in Mansilla Mayor (León) for the year 2011.

Fourier Analysis: Daily Average Temperature							
Harmonics	Outputs	RMSE	R^2^	DW	MPE	FA	AIC
1	T_ave_(t)	2.6686	0.8148	0.4825	0.0266	0.7489	2.6815
2	T_ave_(t)	2.6780	0.8135	0.4794	0.0146	0.7295	2.7047
3	T_ave_(t)	2.7081	0.8093	0.4689	0.0163	0.7357	2.7507
4	T_ave_(t)	2.7035	0.8099	0.4703	0.0145	0.7324	2.7617
5	T_ave_(t)	2.6990	0.8105	0.4721	0.0178	0.7352	2.7712
6	T_ave_(t)	2.7067	0.8095	0.4696	0.0169	0.7337	2.7919
7	T_ave_(t)	2.7183	0.8078	0.4659	0.0190	0.7349	2.8210
8	T_ave_(t)	2.8287	0.7919	0.4320	0.0036	0.7145	2.9505

The best results of the variable (T_ave_(t)) with each statistic (RMSE (°C), R^2^, DW, MPE, FA and AIC) are underlined.

**Table 13 sensors-22-04850-t013:** Effectiveness of the typical annual Fourier function from the 1st to the 8th harmonic of minimum temperature with respect to the SIAR data in Mansilla Mayor (León) for the year 2011.

Fourier Analysis: Daily Minimum Temperature							
Harmonics	Outputs	RMSE	R^2^	DW	MPE	FA	AIC
1	T_min_(t)	3.3747	0.5955	0.5521	0.2576	0.6710	3.3930
2	T_min_(t)	3.3414	0.6035	0.5630	0.2464	0.6733	3.3782
3	T_min_(t)	3.3739	0.5957	0.5521	0.2649	0.6910	3.4296
4	T_min_(t)	3.3942	0.5908	0.5460	0.2881	0.7062	3.4695
5	T_min_(t)	3.3691	0.5969	0.5537	0.2933	0.7511	3.4626
6	T_min_(t)	3.3714	0.5963	0.5530	0.2875	0.7441	3.4839
7	T_min_(t)	3.3738	0.5957	0.5522	0.2652	0.7302	3.5053
8	T_min_(t)	3.4049	0.5883	0.5425	0.2354	0.7002	3.5567

The best results of the variable (T_min_(t)) with each statistic (RMSE (°C), R^2^, DW, MPE, FA and AIC) are underlined.

**Table 14 sensors-22-04850-t014:** SIAR hourly average temperature data in Mansilla Mayor (León) and value resulting from the estimation made with the ASHRAE method for week 10, …, 16 May 2011.

ASHRAE Method: Hourly Average Temperature							
	Outputs	RMSE	R^2^	DW	MPE	FA	AIC
ASHRAE	T_0h_(t), …, T_23h_(t)	2.3586	0.8460	0.2229	0.0948	0.8622	2.4151

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
