# Peer review of "Prediction of Daily Ambient Temperature and Its Hourly Estimation Using Artificial Neural Networks in an Agrometeorological Station in Castile and León, Spain"

_sensors, 2022, doi:10.3390/s22134850_

Round 1
Reviewer 1 Report
In general, the manuscript need to be improved in several critical segments, including proof editing in English. The manuscript is hard to follow, with too many long statements, and Tables that can be moved from the text body to supplements (e.g. Tables 8-10), while the discussion and reference list need to be supported and extended with recent similar studes.
More specifically, the abstract need to be rewritten, with 1-2 finaly conclusing statement. Firs sentence, “this article”, change to this study. Also, last part of the first sentence (in science and technology agricultural), rewrite.
Introduction. First sentence is confused, rewrite.
Line 101, start with: In this study …
Explain with 1-2 sentences what is a novel in your study?
Materials & Method
Line 138, check Mean of means ……
Combine Fig 1 and 2 in one.
Is it possible to move Table 1 to supplements? I would recommend it.
Results & Discussion
Firs statement is too long, need to be split and rewritten.
I think that equations 1a-f and 2a-ccan be moved to supplements.
Discussion section is poorly elaborated and need to be extended with some relevant and recent studies.
For instance, in the references suggested bellow (and their ref. lists) are addressed some of useful data and approaches that can quality improve your discussion.
https://doi.org/10.3390/agronomy12010022
In conclusions continue with using only acronyms, do not spell out full terms which you did already.
Again, in line 522 use: In this study …
Please explain how is possible to use the prediction of ambient temperature in some agricultural applications (e,g, crop water requirements, irrigation), as you stated in lines 520?
Author Response
Comments and Suggestions for Authors
In general, the manuscript need to be improved in several critical segments, including proof editing in English. The manuscript is hard to follow, with too many long statements, and Tables that can be moved from the text body to supplements (e.g. Tables 8-10), while the discussion and reference list need to be supported and extended with recent similar studies.
Authors: Thank you for your suggestions, we have made the appropriate improvements.
More specifically, the abstract need to be rewritten, with 1-2 finaly conclusing statement. Firs sentence, “this article”, change to this study. Also, last part of the first sentence (in science and technology agricultural), rewrite.
Authors: Thank you for your suggestions, we have made the appropriate improvements.
Introduction. First sentence is confused, rewrite.
Authors: Thank you for your suggestions, we have made the appropriate improvements.
Line 101, start with: In this study …
Authors: Thank you for your suggestions, we have made the appropriate improvements.
Explain with 1-2 sentences what is a novel in your study?
Authors: Thank you for your suggestions, we have made the appropriate improvements.
Materials & Method
Line 138, check Mean of means ……
Authors: Thank you for your suggestions, we have made the appropriate improvements.
Combine Fig 1 and 2 in one.
Authors: Thank you for your suggestions, we have made the appropriate improvements.
Is it possible to move Table 1 to supplements? I would recommend it.
Authors: It is possible to move Table 1 to supplements, but instructions for authors of SENSORS journal indicate that all Figures, Schemes and Tables must be inserted in the text near their first citation.
Based on comments of Reviewer 1, we detected some concern about the number and size of Figures and Tables contained in the article. This concern was the same that the authors had when writing the article (Diez et al. 2020: Agronomy 2020, 10, 96; doi: 10.3390/agronomy10010096) also from the MDPI editorial, where solar radiation (daily forecast and hourly estimation) in the same agrometeorological station and of which we have purposely followed the same structure in its writing, in the study presented here, to facilitate future comparisons and the study of other meteorological variables that affect crop production in the same place. The good work of the MDPI editorial solved its layout for a correct reading of the article.
Results & Discussion
Firs statement is too long, need to be split and rewritten.
Authors: Thank you for your suggestions, we have made the appropriate improvements.
I think that equations 1a-f and 2a-c can be moved to supplements.
Authors: It is possible to move Equations 1a-f and 2a-c … 4 to supplements, but in the instructions for authors of SENSORS journal, it is indicated that everything that is inserted in the main text should be done close to its first citation. Furthermore, Equations 2a-c … 4 are results of the classical models evaluated.
As we have done previously, we ask Reviewer 1 for confidence in this regard, given the good reception that the article (Diez et al. 2020: Agronomy 2020, 10, 96; doi: 10.3390/agronomy10010096) has have, from where we follow the same structure of this article.
To improve the readability of the equations we have used the normal numbering formula without adding letters, at the request of Reviewer 2.
Discussion section is poorly elaborated and need to be extended with some relevant and recent studies. For instance, in the references suggested bellow (and their ref. lists) are addressed some of useful data and approaches that can quality improve your discussion. https://doi.org/10.3390/agronomy12010022
Authors: Thank you for your suggestions, we have made the appropriate improvements.
In Conclusions continue with using only acronyms, do not spell out full terms which you did already.
Authors: Thank you for your suggestions, we have made the appropriate improvements.
Again, in line 522 use: In this study …
Authors: Thank you for your suggestions, we have made the appropriate improvements.
Please explain how is possible to use the prediction of ambient temperature in some agricultural applications (e,g, crop water requirements, irrigation), as you stated in lines 520?
Authors: Thank you for your suggestions, we have made the appropriate improvements.
Reviewer 2 Report
One disadvantage of the publication is that it is narrowed down to one weather station, maybe it would be worth extending the measurement data to southern Spain? It is worth describing the substantive justification for using this model for various agricultural crops in this region.
Line 101-112:“ …using the least possible number of inputs, in order to facilitate its practical application in the processes of irrigation needs and evaluation systems, or to predict plant growth, or disease control.” Please cite the literature as to whether we can achieve these goals by monitoring and predicting temperatures.
It would be worth making and presenting an example using a model for agriculture, e.g. with predicting unfavorable temperatures associated with frosts for a specific agricultural crop, then the publication would be much more interesting and practical.
Optionally, this publication could also include humidity, which is found in future research plans (page 20, line 542-543).
A nomenclature with a description of the quantities and units should be added to the publication.
It would be more readable to use the normal formula numbering without adding letters.
Author Response
Comments and Suggestions for Authors
One disadvantage of the publication is that it is narrowed down to one weather station, maybe it would be worth extending the measurement data to southern Spain? It is worth describing the substantive justification for using this model for various agricultural crops in this region.
Authors: This article that we have sent to the journal Sensors and our previous article published in the journal Agronomy (Diez et al. 2020: Agronomy 2020, 10, 96; doi: 10.3390/agronomy10010096), come from the Doctoral Thesis carried out by one of the authors, Francisco J. Diez, where, among other topics, deals with the daily prediction and hourly estimation of meteorological variables (global solar irradiation and ambient temperature).
The study of the prediction of the global solar irradiation on the horizontal plane was presented at the X Iberian Congress of Agroengineering, from where its publication in the journal Agronomy was recommended. The currect study of the prediction of the daily ambient temperature (maximum, average and minimum) and its hourly estimation was presented at the XI Iberian Congress of Agroengineering, from where its publication in the journal Sensors was recommended, for which you have been assigned as Reviewer 2.
Both articles study meteorological variables recorded by the same agrometeorological station, but in the case of the article published in Agronomy, the global solar irradiation on the horizontal plane of the place is studied, and in the case of the current article sent to Sensors, another different meteorological variable, maximum, average and minimum ambient temperatures, are studied. Due to the acceptance of the first article published in the journal Agronomy, we have tried in this new article to follow the same structure in both articles, with the aim of facilitating future comparisons.
Responding to the proposal of Reviewer 2, to extend the study to data from the south of Spain, there would be no problem, but in this article a methodology that can be applied to any place is presented, so the scope of application is that of the place of the data used for the training of the ANNs, and this is where their competitive advantage lies. The data from one place to another, although close, have variations. When creating an ANN model with data from a specific location, we are giving the model the knowledge acquired in that place and by means of the theory of the ANNs, they are in charge of finding the hidden relationships.
Line 101-112:“ …using the least possible number of inputs, in order to facilitate its practical application in the processes of irrigation needs and evaluation systems, or to predict plant growth, or disease control.” Please cite the literature as to whether we can achieve these goals by monitoring and predicting temperatures.
Authors: Thank you for your suggestions, we have made the appropriate improvements. We have introduced 4 new references.
It would be worth making and presenting an example using a model for agriculture, e.g. with predicting unfavorable temperatures associated with frosts for a specific agricultural crop, then the publication would be much more interesting and practical.
Authors: Thank you for your suggestions, we have made the appropriate improvements. We have introduced three crop evapotranspiration models for estimating irrigation needs for presenting an example using a model for agriculture (see Conclusions section).
Optionally, this publication could also include humidity, which is found in future research plans (page 20, line 542-543).
Authors: It is our intention in future research works to study the models for the humidity variable and for other meteorological variables, as cited in the Conclusions section
A nomenclature with a description of the quantities and units should be added to the publication.
Authors: Thank you for your suggestions, we have made the appropriate improvements. We have introduce the Nomenclature section
It would be more readable to use the normal formula numbering without adding letters.
Authors: Thank you for your suggestions, we have made the appropriate improvements.
Reviewer 3 Report
- Between lines of 109-112, the text included seems not part of the objectives, it seems part of the results. This part should be reviewed.
- Between the lines 254-258, is missing the presicion.
- The lines of research proposed in the lines 542-548, it seems is not for the audience concern as so far, in above already were mentioned the rest of variables.
Author Response
Comments and Suggestions for Authors
Between lines of 109-112, the text included seems not part of the objectives, it seems part of the results. This part should be reviewed.
Authors: The last paragraph of the Introduction section, to which Reviewer 3 refers, has been made, "intentionally", by the authors as a small summary, to place the reader of the article in all its development (i.e., introduction to the study carried out, materials used and methods developed, main objective of the work, main results obtained and their novelty, as suggested by Reviewer 1). It is a practice of the authors, which we also use in other published articles and which we understand improves the contextualization of the work.
Between the lines 254-258, is missing the precision.
Authors: Regarding the lack of precision referred to by Reviewer 3, the authors have doubts whether it refers to the results of the daily and hourly predictions, or to the measured data used that are recorded in the SIAR agrometeorological station. If the precision refers to the predictions, the precision values are found in the following sections 3.1. ANN models and 3.2. Classic Models. If the accuracy refers to the data measured at the SIAR agrometeorological station, the measurement ranges of the temperature sensor used are found in section 2.1. Ambient Temperature Data.
The lines of research proposed in the lines 542-548, it seems is not for the audience concern as so far, in above already were mentioned the rest of variables.
Authors: The short-term prediction of temperature and other meteorological variables are of great importance in agriculture, in order to predict agricultural processes of great interest, such as irrigation needs, production forecasts or the appearance of pests and diseases. Although temperature is the main variable of influence, others are also of interest, such as incident solar radiation (Diez et al. 2020: Agronomy 2020, 10, 96; doi:10.3390/agronomy10010096), ambient humidity, cloudiness or precipitation). Therefore, the proposal to add other meteorological variables to improve the performance of the ANN prediction models proposed here is of interest in agricultural modelling and is in accordance with the theory of time series using artificial intelligence.
Round 2
Reviewer 1 Report
The authors have not activated track change option, so it is not possible to see all modefications in the R1, but seems that a litlle of suggestions have implemented.
In general, the manuscript still need to be improved in several critical segments, including proof editing in English. The manuscript is hard to follow, with too many long statements, and Tables that can be moved from the text body to supplements (e.g. Tables 8-10), while the discussion and reference list need to be supported and extended with recent similar studes.
More specifically, the abstract need to be rewritten, with 1-2 finaly conclusing statement. Firs sentence, “this article”, change to this study. Also, last part of the first sentence (in science and technology agricultural), rewrite.
Introduction. First sentence is confused, rewrite.
Line 101, start with: In this study …
Materials & Method
Line 138, check Mean of means ……
Combine Fig 1 and 2 in one.
Is it possible to move Table 1 to supplements? I would recommend it.
Results & Discussion
Firs statement is too long, need to be split and rewritten.
I think that equations 1a-f and 2a-ccan be moved to supplements.
Discussion section is poorly elaborated and need to be extended with some relevant and recent studies.
For instance, in the reference suggested bellow (and its ref. lists) are addressed some of useful data and approaches that can quality improve your discussion.
https://doi.org/10.3390/agronomy12010022
In conclusions continue with using only acronyms, do not spell out full terms which you did already.
Again, in line 522 use: In this study …
Please explain how is possible to use the prediction of ambient temperature in some agricultural applications (e,g, crop water requirements, irrigation), as you stated in lines 520?
Reviewer 2 Report
My comments were clarified.